# Inherited Proteoglycan Biosynthesis Defects—Current Laboratory Tools and Bikunin as a Promising Blood Biomarker

**DOI:** 10.3390/genes12111654

**Published:** 2021-10-20

**Authors:** Walid Haouari, Johanne Dubail, Christian Poüs, Valérie Cormier-Daire, Arnaud Bruneel

**Affiliations:** 1INSERM UMR1193, Paris-Saclay University, Faculté de Pharmacie, 5 rue Jean-Baptiste Clément, 92220 Châtenay-Malabry, France; walid.haouari@universite-paris-saclay.fr (W.H.); christian.pous@u-psud.fr (C.P.); 2INSERM UMR1163, French Reference Center for Skeletal Dysplasia, Imagine Institute, Paris University, 24 Boulevard du Montparnasse, 75015 Paris, France; johanne.dubail@inserm.fr (J.D.); valerie.cormier-daire@inserm.fr (V.C.-D.); 3AP-HP, Necker Enfants Malades Hospital, 149 rue de Sèvres, 75015 Paris, France; 4AP-HP, Biochimie Métabolique et Cellulaire, Hôpital Bichat-Claude Bernard, 46 rue Henri Huchard, 75018 Paris, France

**Keywords:** bikunin, CDG, linkeropathies, GAG, proteoglycans, skeletal dysplasia

## Abstract

Proteoglycans consist of proteins linked to sulfated glycosaminoglycan chains. They constitute a family of macromolecules mainly involved in the architecture of organs and tissues as major components of extracellular matrices. Some proteoglycans also act as signaling molecules involved in inflammatory response as well as cell proliferation, adhesion, and differentiation. Inborn errors of proteoglycan metabolism are a group of orphan diseases with severe and irreversible skeletal abnormalities associated with multiorgan impairments. Identifying the gene variants that cause these pathologies proves to be difficult because of unspecific clinical symptoms, hardly accessible functional laboratory tests, and a lack of convenient blood biomarkers. In this review, we summarize the molecular pathways of proteoglycan biosynthesis, the associated inherited syndromes, and the related biochemical screening techniques, and we focus especially on a circulating proteoglycan called bikunin and on its potential as a new biomarker of these diseases.

## 1. Introduction

Proteoglycans (PGs) consist of core proteins linked to sulfated glycosaminoglycan (GAG) chains. They constitute a family of around fifty macromolecules involved in a wide variety of pathophysiological processes in humans [1]. They confer the biomechanical properties of the osteoarticular system and of the connective tissues of virtually all the organs during fetal development, growth, and aging. PGs are hydrophilic compounds mostly present in the extracellular matrices (ECM) in which they interact with each other and with hyaluronan and collagens. Such interactions are fundamental for the organization of the ECM and its viscoelastic properties [2]. PGs also mediate cell signaling through their binding to various ligands such as microbial pathogens [3], cytokines [4], and growth factors [5]. Hence, they modulate inflammatory and healing processes by promoting cell recruitment and proliferation during infections, cancers, and wound repair (Appendix A).

PG biosynthesis occurs in the endoplasmic reticulum (ER) and in the Golgi apparatus and involves the interplay between numerous molecular actors along the secretory pathway. Inherited mutations causing defects in this system could result in severe orphan diseases called “PG inherited metabolic disorders” (PG-IMD). The causative variants occur in genes that encode enzymes responsible for the elongation and modifications of the GAG chains. They may also affect the proteins and transporters that regulate the synthesis and delivery of GAG assembly substrates. PG-IMD share osteoarticular malformations and are often associated with cutaneous abnormalities, heart defects, neurological disorders, deafness, cataracts, and tooth abnormalities [6,7,8]. Otherwise, mucopolysaccharidoses (MPS), a group of diseases resulting from defective GAG catabolism, have very similar clinical presentations to PG-IMD. They are caused by pathogenic variants in genes encoding lysosomal GAG-degrading enzymes leading to deleterious GAG accumulation in tissues [9].

Regarding the highly variable and unspecific symptomatology of PG-IMD, convenient biochemical tests are needed to orient towards PG-IMD-causing gene variants or to ascertain the causality of the mutations unveiled by whole exome sequencing. Whatever the diagnosis route in PG-IMD, the current laboratory assays combine genetic sequencing (gene panels or whole exome sequencing) with the analysis of PG content in patient-derived fibroblasts using substrate radiolabeling, chromatography, mass spectrometry (MS), and Western blotting. Recently, a serum PG called bikunin (Bkn) has emerged as a potential new biomarker allowing the rapid detection of several PG biosynthesis deficiencies [10]. Here, we present an overview of the structure of PGs, their biosynthetic pathways, the related inherited diseases, and the current laboratory screening techniques. Furthermore, we highlight the potentials and discuss the limits of serum Bkn as a new versatile biomarker for the screening and diagnosis of PG-IMD.

## 2. Structure, Synthesis, and Modifications of Proteoglycans

As detailed in Figure 1, PGs are composed of a core protein linked to one or several highly sulfated glycosaminoglycan (GAG) chains such as chondroitin sulfate (CS), dermatan sulfate (DS), heparan sulfate (HS), heparin, or keratan sulfate (KS). Except for KSPGs, the GAG chains are linked to serine (Ser) residues of the core proteins through a common tetrasaccharide linker motif [(glucuronic acid–galactose–galactose–xylose) abbreviated as (GlcA–Gal–Gal–Xyl)]. GAG chains can then be differentiated according to distinct repeated disaccharide motifs linked to the common tetrasaccharide, i.e., [GlcA–GalNAc] for CS, [IdoA–GalNAc] for DS, [GlcA–GlcNAc] for HS, and [IdoA–GlcN] for heparin (list of abbreviations page 13). Regarding KS, they are composed of a [GlcNAc–Gal]_n_ backbone with three types of linkage to the core protein (Figure 1). Furthermore, several sulfate groups are branched on the sugar moieties of the GAG chains, making PGs very negatively charged, which is important for their biological functions [11].

The synthesis of the initiating tetrasaccharide linkage region starts by a serine-O-xylosylation catalyzed by xylosyltransferases (XYLT1 or XYLT2). This first reaction begins in the ER exit sites or in ER/Golgi intermediate compartments and is achieved in the *cis*-Golgi. The two following reactions, which also occur in the *cis*-Golgi, are the sequential additions of two Gal residues by the galactosyltransferases B4GALT7 and B3GALT6, respectively [12,13]. Afterwards, a GlcA is added by the glucuronyltransferase B3GAT3 in the medial-Golgi [14]. Noticeably, after the addition of the first Gal catalyzed by B4GALT7, the serine-linked Xyl residue is 2-*O* phosphorylated by FAM20B kinase (Figure 2). This phosphorylation is mandatory since it allows the substrate recognition by B3GALT6 and B3GAT3 for the addition of the two subsequent monosaccharides [15]. The Xyl-phosphorylation is transient since the phosphate group is removed by XYLP phosphatase after the achievement of tetrasaccharide synthesis [16]. Additionally, in CS-PGs and DS-PGs, a sulfation occurs either on the first or the second Gal residue of the linker region and may have a regulatory role for the following GAG elongation [17,18].

After the formation of the tetrasaccharide linker, the GAG chain elongation starts with the addition of the first *N*-acetyl aminosugar (i.e., GalNAc for CS/DS or GlcNAc for HS/heparin) (Figure 1 and Figure 2). Thereafter, stepwise enzymatic additions of n-repeated disaccharide motifs produce the backbone of the GAG chains. For CSPGs, the *N*-acetygalactosaminyl-transferase CSGALNACT1 or CSGALNACT2 transfers a GalNAc on the terminal GlcA of the linkage region [19]. The following polymerization of the [GlcA–GalNAc]_n_ backbone is ensured by a chondroitin sulfate synthase complex (CHSY) composed of six enzymes, namely CHSY1, CHSY2, CHSY3, chondroitin polymerizing factor (CHPF), CSGALNACT1, and CSGALNACT2 [20,21,22,23]. A family of glycosyltransferases called exostosins (EXT) and exostosin-like (EXTL) proteins mediates HS elongation. The GlcNAc transfer to the linkage region is catalyzed by EXTL2 or EXTL3 and the following [GlcA–GlcNAc]_n_ polymerization is ensured by EXTL1, EXT1, and EXT2 [24,25,26] (Figure 1). In type-I KS, the elongation starts with the N-glycan branching of a GlcNAc residue to an asparagine (Asn) of the core protein. In type-II KS, a GalNAc is added to a Ser/Thr to form a mucin core-2 *O*-glycosidic linkage. Regarding type-III KS, the linkage region consists of a mannose *O*-linked to a Ser/Thr of the core protein. The three KS GAG-type chains consist of a sulfated [GlcNAc–Gal]_n_ (polylactosamine) backbone that is synthesized by B3GlcNAcT7, B4GALT4, and the sulfotransferase KSGal6ST [27,28,29].

Various modifications, notably including sulfation reactions, occur on the repeated disaccharide motifs of the GAG chains (Figure 1). For the CS chains, chondroitin-4 sulfotransferases (C4ST)-1, -2, and -3 as well as C6ST-1 and -2 mediate the sulfation of several GalNAc residues [30,31,32,33]. GlcA residues can also be sulfated by the uronic acid sulfotransferase (U2ST) [34]. Furthermore, varying proportions of GlcA residues of the CS chains can be converted to IdoA by the DS epimerases (DSE)-1 and -2 to form DS chains [35,36]. Sulfations occur on both GalNAc and IdoA residues via D4ST1 (CHST14) and U2ST, respectively [34,35,36,37]. For HSPGs, several GlcNAc residues undergo *N*-deacetylation and subsequent *N*-sulfations catalyzed by bifunctional enzymes named GlcNAc *N*-deacetylase/*N*-sulfotransferases (NDSTs) [38,39]. The resulting sulfoglucosamine (GlcNS) residues can undergo an additional 6-*O* sulfation via a 6-*O* sulfotransferase to form GlcNS,6S [40]. Additionally, for the formation of heparin, some GlcA residues are converted by the HS-epimerase to IdoA residues, which are further sulfated [41]. For KSPGs, sulfations occur on both GalNAc and Gal residues composing the KS chain and are catalyzed by Gal6ST (CHST1) and GlcNAc6ST (CHST5 and/or CHST6) [42,43,44]. Finally, minor modifications of some GAG chains by sialic acid and fucose residues have been reported [45].

PG biosynthesis enzymes are transmembrane proteins having precise localization into the successive subcompartments of the Golgi apparatus according to the reaction they catalyze. This organization is maintained by a balance between the anterograde and retrograde Golgi vesicular trafficking [46]. While the molecular motors dyneins and kinesins transport cargo vesicles along microtubules, the coat protein complexes (COP 1 and COP 2) and tethering factors such as the conserved oligomeric complex (COG 1–8) participate in vesicle budding and fusion [47].

The substrates of glycosyltransferases during GAG elongation are uridine-diphosphate (UDP)-sugars. Their synthesis occurs in the cytosol and involves numerous metabolic pathways including the formation of phospho-sugars (e.g., Gal-1-P, GalNAc-1-P) and a process frequently called “activation” consisting of the transfer of UDP to the phospho-sugars by specific UDP-sugar pyrophosphorylases [48,49]. Afterwards, the activated UDP-sugars are translocated into the lumen of the ER and the Golgi apparatus through transmembrane nucleotide-sugar transporters (NSTs). At least 31 NSTs belonging to the solute carrier 35 family (SLC35) have been described [50,51]. Glycosylation reactions release free UDP molecules, which are further converted by nucleotidases such as calcium-activated nucleotidase-1 (CANT1) to uridine-monophosphate [52], the latter being exported to the cytosol by NSTs to the benefit of new UDP-sugar molecules.

For sulfation reactions, sulfotransferases use the sulfate donor 3′-phospho-adenosine 5′-phospho-sulfate (PAPS). The biosynthesis of PAPS occurs in the cytosol and is mediated by PAPS synthases (PAPSS 1 and 2) that catalyze the transfer of phospho-adenosine-phosphate to inorganic sulfate (SO_4_^2−^) [53]. The activated PAPS molecules are then transported into the Golgi lumen by SLC35B2 and SLC35B3 transporters [54,55]. GAG sulfation reactions release free PAP that is hydrolyzed by inositol-monophosphatase 1 (IMPAD1) into adenosine-monophosphate and inorganic phosphate [56].

## 3. Classification, Distribution, and Roles of Proteoglycans

PGs can be classified into four classes (Appendix A) according to their cellular localization. The unique known intracellular PG is serglycin, which carries a heparin chain or a CS chain according to the cell type. It is involved in the storage of histamine and proteases into secretory granules in mast cells and macrophages for their delivery during inflammatory response [57].

The second class of PGs includes those embedded into the cell plasma membrane such as syndecans and glypicans. In numerous cell types including leucocytes and epithelial cells, their extracellular domain binds several ligands such as the vascular endothelial growth factor (VEGF) and inflammatory cytokines. This leads to various signal transduction pathways regulating leukocyte recruitment, angiogenesis, cell proliferation, and differentiation [58]. They also interact with ECM components such as collagens, hyaluronan, and fibronectin to form the ECM network [59]. Otherwise, the cytoplasmic domain of cell surface PGs interacts with the cytoskeleton, thereby participating in cell motility [60].

Pericellular PGs (the third class) are linked to the cell surface through adhesion molecules such as integrins and constitute a supportive matrix allowing interactions with the cellular microenvironment. For example, perlecan is a ubiquitous HSPG located in the basement membranes, which is involved in healing and angiogenesis during wound repair by interacting with collagen IV and VEGF, respectively [61].

The fourth class of PGs comprises the extracellular PGs that are composed of hyaluronan- and lectin-binding PGs (hyalectans) and the small leucin-rich proteoglycans (SLRPs). Aggrecan is a hyalectan that carries several CS, DS, and KS GAG chains and is the main PG of cartilage. It forms bulky aggregates with hyaluronan, collagen fibrils, and with other PGs to generate a hydrated gel underlying the viscoelastic consistence and the resistance of cartilage [62]. Biglycan is a CS/DS PG of the SLRPs family that is ubiquitously found in ECMs. Among the wide range of its physiological functions, biglycan is involved in bone formation by regulating osteoblast differentiation through interaction with the bone morphogenic protein (BMP) [63]. Decorin, an SLRP carrying a DS chain, is mostly present in skin ECM where it interacts with collagen I, epidermal growth factor receptor, and FGF to promote wound healing and to attenuate tumor growth [64]. In blood and urine, the inter-α-trypsin inhibitor proteins (IAIPs) are SLRPs carrying a unique short CS chain branched on the core protein Bkn. Their roles are described below in a dedicated section. Another circulating PG called endocan is a DSPG secreted by endothelial cells that inhibits leucocyte migration and adhesion to inflamed tissues as well as angiogenesis and cell proliferation during wound healing and tumor progression [65] (Appendix A).

## 4. Inborn Errors of Proteoglycan Metabolism

PG-IMD mostly consist of skeletal, connective tissue, and cartilage defects. The phenotypes comprise various osteoarticular manifestations including skeletal dysplasia, joint dislocations, and dysmorphisms [6]. Patients also display other symptoms including skin hyperelasticity, neurological disorders, growth retardation, heart defects, deafness, tooth abnormalities, and ocular troubles. PG-IMD can be classified according to the defective step of PG biosynthesis. Therefore, one can distinguish those caused by impaired tetrasaccharide linker formation (linkeropathies), GAG elongation, GAG sulfation, substrate supply, and by defects in Golgi homeostasis (including Golgi vesicular trafficking and ionic environment) (Appendix A). Importantly, it appears that no clinical characteristic could differentiate a PG-IMD subgroup from another one.

Linkeropathies designate disorders due to the defective synthesis of the common tetrasaccharide linkage region. They arise from pathogenic variants in XYLT1, XYLT2, B4GALT7, B3GALT6, and B3GAT3, which result in Desbuquois dysplasia, spondylo-ocular syndrome, Ehlers–Danlos type 1 and type 2, and Larsen-like syndrome, respectively [12,66,67,68,69]. Otherwise, benign bone tumors called exostosis are displayed during inherited defects of HS polymerizing enzymes (i.e., EXT1 and EXT2), the resulting “hereditary multiple exostosis syndrome” (HMES) being the most frequent skeletal genetic disorder with a prevalence estimated at 1:50,000 [70,71]. GAG sulfation defects, which arise from defective sulfate uptake, activation, or linkage to the GAG chains (Appendix A), lead to a range of unspecific skeletal disorders including severe achondrogenesis and spondylo-epiphyseal dysplasia [72]. Most pathogenic variants in genes encoding sugar transporters and Golgi homeostasis regulators lead to congenital disorders of glycosylation (CDG) [73]. Affected individuals mostly display impaired N- and O-glycosylation of proteins, but additional alterations of the PG metabolism have been reported in SLC35A2-CDG and SLC35A3-CDG (i.e., UDP-Gal and UDP-GlcNAc Golgi transporter defects) [74,75], in deficiencies of COG4 and GORAB (i.e., proteins regulating the Golgi retrograde trafficking) [76,77], and in TMEM165-CDG and SLC10A7-CDG (i.e., Mn^2+^ and Ca^2+^-related transporter defects, respectively) [78,79]. The clinical manifestations of these CDG with PG defects remain wide and unspecific like in other CDG. Noticeably, arthrogryposis and cutis laxa are displayed by SLC35A3-CDG and GORAB-CDG, respectively, while very severe skeletal dysplasia is associated with major osteoporosis and tooth abnormalities during TMEM165 and SLC10A7 deficiencies, respectively. Finally, a condition with severe developmental delay and epileptic seizures has been described in individuals with pathogenic variants in UGDH, a gene coding for UDP-glucose dehydrogenase that forms UDP-GlcA from UDP-glucose [80] (Appendix A).

## 5. Current Laboratory Tools and Examples of Applications

The low incidence of PG-IMD and their unspecific clinical signs make a diagnosis only based on a clinical approach difficult. As shown in Figure 3, the screening strategy benefits from the development of next-generation sequencing techniques (gene panels, whole exome/genome sequencing) classically applied to DNA from blood-derived lymphocytes for the early identification of potentially causative gene variants. However, to ascertain their pathogenicity and perform an accurate diagnosis, the implementation of tedious biochemical assays is often required. In most cases, invasive skin biopsies are performed and addressed to specialized laboratories where the overall PG content of fibroblasts is evaluated and compared to controls. The most common strategy involves fibroblasts cell culture in the presence of radiolabeled substrates (^35^S and ^3^H-radiolabelled sugars) that enter the cells and are incorporated to the GAG chains of PGs. Afterwards, liquid scintillation counting is employed to quantify labeled GAGs in cell culture media or lysates (Figure 3). For example, this technique allowed the detection of decreased PG biosynthesis in XYLT1- and CANT1-deficient patient fibroblasts [66,81]. The quantification of the fibroblast PG content by immunostaining is also possible using antibodies and/or lectins recognizing the GAG chains. Enzyme-linked immunosorbent assay using anti-CS/DS antibodies showed reduced immunostaining in CSGALNACTl-deficient patient fibroblasts [82]. Flow cytometry using anti-CS and -HS antibodies showed a significant decrease in cell surface CS and HSPG in patients with B3GAT3 linkeropathy [69]. More accurate analyses allowing the determination of GAG disaccharide composition and sulfation degree could be performed using high-performance liquid chromatography (HPLC) and mass spectrometry (MS) following specific enzymatic treatments (Figure 3). Indeed, the disaccharide units of CS/DS, HS/heparin, and KS GAG chains are obtained by treating the cell lysates with chondroitinase, heparitinase, and keratanase (i.e., GAG lyases), respectively. HPLC analysis of chondroitinase-treated fibroblast lysates from DSE-deficient patients showed decreased amounts of [IdoA–GalNAc] disaccharide (i.e., the DS backbone) compared to controls [83]. HPLC–MS coupling highlighted reduced [IdoA-GalNAc4-*O*-sulfate] levels in chondroitinase-treated cell fractions from D4ST1-deficient patients compared to controls [84]. Otherwise, specific PGs from skin fibroblasts such as decorin and biglycan can be used as markers, using antibody-based Western blot techniques, to highlight defective PG biosynthesis, as it has been performed for a patient with B4GALT7 linkeropathy [85].

Fibroblasts are also convenient cells for functional biochemical analyses to highlight loss of function resulting from the suspected gene variant. Indeed, decreased enzymatic activities were shown for XYLT2 [67], B4GALT7 [85], and B3GAT3 [69] in patient fibroblasts. In the case of defective UDP-sugar transporters, substrate uptake assays can be performed as shown in SLC35A3 mutated patient fibroblasts with reduced UDP-GlcNAc transport [86]. Additionally, observations of the Golgi apparatus using electron and/or fluorescence microscopy can highlight morphological alterations, impaired anterograde/retrograde trafficking balance, and the mislocalization of enzymes and transporters involved in PG biosynthesis such as in COG4 deficiency [76] (Figure 3). Less commonly, blood and urine can be used to quantify total GAG levels. Purification steps involving anion exchange chromatography and desalting allow the isolation of GAGs, which are further digested using GAG lyases (Figure 3). GAG disaccharide composition and sulfation degree can then be analyzed using HPLC or MS as it has been performed for patients with EXT1 and EXT2 deficiency where HS-related disaccharide plasma levels were decreased compared to controls [87]. Blood and urine samples can also be already treated with GAG lyases and applied to a centrifugal filter before analyzing GAG disaccharide containing filtrate by HPLC/MS. This technique showed decreased HS serum levels in EXTL3-deficient cases [88], while major hyposulfation of CS chains was detected in urine from one *CHST3* mutated patient [89]. Regarding MPS, their biochemical diagnosis is mostly based on a colorimetric method using dimethyl-methylene blue, which switches to purple when mixed with urine samples containing increased GAG concentrations. Qualitative GAG analyses from blood and urine are also performed using HPLC/MS to determine the defective GAG catabolic step [90].

The above biochemical techniques are time-consuming and often require complex technical handling and expensive material. Skin biopsies are invasive, especially for young children. Moreover, in vitro primary culture of skin fibroblasts can trigger phenotype changes and bias or affect the reproducibility of the results. Additionally, cell cultures are vulnerable to microbial contaminations and thus need skilled personnel able to ensure cautious handling for long-term cell viability. Blood and urine constitute more affordable biological material with easy sampling and storage. However, the complexity of blood composition and the presence of high amounts of salts in urine can hinder the following HPLC and/or MS analyses of the PG and/or GAG content. Therefore, some preliminary purification and desalting steps are frequently required [87].

Regarding the current lack of rapid and simple routine analyses for the screening/diagnosis of PG biosynthesis defects, the discovery and development of new blood and/or urine biomarkers are mandatory. In this context, we recently evaluated a new biochemical test based on the analysis of serum bikunin (Bkn), a CS-type PG found in the blood at high levels [10,91]. In the following sections, we describe the structure and roles of this proteoglycan and discuss its potentials and limits as a convenient blood biomarker of PG-IMD.

## 6. Bikunin Proteoglycan Isoforms

Bikunin (Bkn) is a serum protein mainly synthesized by the liver. It constitutes the core protein of a group of original CSPGs known as the inter-α-trypsin inhibitor proteins (IAIPs) (Figure 4). They consist of three isoforms, i.e., one light form (35 to 45 kDa) corresponding to the core Bkn linked to a short CS chain (Bkn–CS), and two heavy forms named Pro-α-trypsin inhibitor (PαI) (125 kDa) and Inter-α-trypsin inhibitor (ITI) (225 kDa). In ITI, the CS chain is esterified by two different heavy chain (HC1 or HC2) glycoproteins, while HC3 attachment produces PαI (Figure 4) [92]. CS chain elongation on the core protein involves the classical pathway of CSPG biosynthesis (described above) with the linkage of the HCs to a distal GalNAc residue in the *trans*-Golgi network (TGN). Such a “protein-GAG-protein(s)” structure has been described exclusively in IAIPs [93]. The produced isoforms are then secreted into the blood at high levels, with ITI and PαI being the major circulating isoforms [94]. In urine, no heavy form is normally found since they do not cross the glomerular barrier, but Bkn–CS is present in rather high amounts [95]. Note that IAIP serum and urine levels could be markedly modulated in pathological situations such as cancer, inflammation, liver diseases, and infectious diseases [96,97].

Concerning the physiological roles of IAIPs, they can be extravasated under various stimuli and then can exchange their HCs with hyaluronan to form HC–hyaluronan complexes that stabilize the ECM [98]. Such an exchange has protective effects against the degradation of joint ECM during osteoarthritis [99]. It also promotes the sequestration and inactivation of leucocytes during sepsis [100] and the protection of oocytes and of the amniotic membrane during ovulation and fetal development [101]. Additionally, the Bkn core protein carries inhibitory activity towards inflammation-associated proteases [102]. Otherwise, HCs have been shown to interact with vitronectin, a glycoprotein of the ECM involved in tissue repair [103].

## 7. Serum Bikunin Analyses in PG-IMD

The high serum levels of Bkn isoforms together with the availability of specific antibodies enable their easy, rapid, and rather cost-effective Western blot analysis for the screening of PG-IMD. Furthermore, Bkn–CS has a simple structure consisting of a unique and short CS chain branched on a small core protein not subjected to marked polymorphisms. These features facilitate the Western blot profile reading and interpretation by highlighting marked differences compared to controls in PG defects.

Western blot analysis of serum protein samples allows the detection of the three Bkn isoforms (ITI, PαI, and Bkn–CS) as well as the free Bkn core protein, with ITI and PαI being the major isoforms [10,91]. Analyses in dried blood spots and in urine have also been considered and gave encouraging results (unpublished data). In complement to classical Western blot, two-dimensional electrophoresis (2-DE) of Bkn–CS can highlight the negative charges provided by sulfate and phosphate groups [10]. However, it must be noted that 2-DE does not provide accurate structural data on GAG chains in contrast to mass spectrometry. Moreover, 2-DE technical handling could be rather delicate for inexperienced operators.

We addressed in Figure 5 the stepwise biosynthesis of Bkn–CS within the secretory pathway and highlighted the respective involvements of known PG-IMD-associated genes. This schematic overview points out the high diversity of the metabolic pathways engaged towards GAG elongation as well as the importance of their harmonious interplay. We also highlighted in Figure 5 the genes whose pathogenic variants impaired Bkn–CS biosynthesis and those for which Bkn analysis has not been performed yet, with anticipated potential results. We showed abnormal tetrasaccharide formation in patients with B4GALT7, B3GALT6, and B3GAT3 linkeropathies and impaired CS elongation in the CHSY1 inherited defect, with the ability to discriminate between respective deficiencies using 2-DE [10,91]. For the FAM20B kinase deficiency, it is likely that the 2-DE of serum Bkn–CS would reveal a lack of xylose phosphorylation.

In PG-IMD due to defective sugar transport and activation, we showed abnormalities in one individual with SLC35A3-CDG (impaired UDP-GlcNAc transporter) but a normal profile in one SLC35A2-CDG case (impaired UDP-Gal transporter), suggesting, at least in this individual, a preferential supply of UDP-Gal to GAG elongation rather than N-glycosylation. It would be interesting to analyze serum Bkn in CANT1 and SLC35D1 deficiencies. Otherwise, the Bkn–CS profile was clearly abnormal in TMEM165-CDG, and it would be mandatory to perform the analysis of other PG-IMD with Golgi homeostasis impairments such as COG4 and GORAB deficiencies. Finally, Bkn–CS sulfation can be assessed, notably using 2-DE, with interesting potentials in PG-IMD with impaired sulfotransferases (CHSTs), sulfate transporters (SLC26A2, SLC35B2), PAPS synthases (PAPSS1/2), and IMPAD (Figure 5). This way, we recently found altered Bkn–CS profiles in one SLC35B2-deficient individual (submitted publication).

Altogether, these features show that the Bkn–CS profile could bring a rapid evaluation of the overall functionality of PG biosynthesis in a simple screening laboratory tool. Moreover, the Bkn signature profiles displayed in some PG-IMD [10,91] suggested the ability to provide an accurate identification of some causative gene deficiencies. Nevertheless, serum Bkn–CS analysis is limited to the screening of pathological gene variants expressed by the liver. Accordingly, we observed a normal profile in XYLT1 linkeropathy (XYLT1 is not expressed in the liver). Likewise, the Bkn–CS profile was normal in a patient with the CHST3 pathogenic variant (personal data) and should also be normal in CSGALNACT1 and U2ST inherited deficiencies since the liver does not express these genes. Furthermore, during specific alterations of HS or KS GAG elongation and sulfation, it is very likely that Bkn–CS may be unaffected, which remains to be tested in EXT1/2, EXTL2, CHST6, and CHST14 mutated individuals (Figure 5). Note that an isolated HS biosynthesis defect was shown during SLC10A7 deficiency [80]; the latter could therefore lead to an unaffected Bkn–CS profile. Evaluating additional circulating PGs such as PG-100, endocan, or apolipoprotein-O could compensate such limitations of Bkn–CS isoforms as PG-IMD blood biomarkers.

## 8. Conclusions

The biosynthesis of PGs constitutes an intricate network of diverse metabolic pathways whose study proves to be a daunting task. Moreover, the severe clinical consequences of PG-IMD emphasize the major importance of PGs in pathophysiology and recall the need for improving the monitoring of these disorders. So far, the biochemical screening strategy relies on rather laborious technical implementations with a lack of routine assays to alleviate the burden of diagnosis for specialized clinicians and biologists. In this context, a single Bkn–CS isoform analysis is a promising tool that can direct the diagnosis towards numerous PG-IMD simultaneously, at least for liver-expressed gene mutants.

## Figures and Tables

**Figure 1 genes-12-01654-f001:**
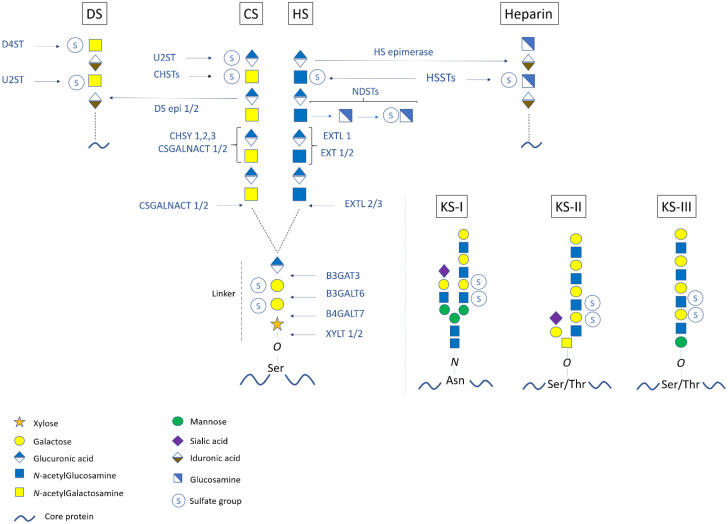
Biosynthesis and modifications of proteoglycans. Upon the core protein-linked tetrasaccharide linkage region (i.e., GlcA–Gal–Gal–Xyl), GAG chain polymerization starts with the transfer of a first amino sugar (i.e., GalNAc for CS and GlcNAc for HS) and continues with the addition of repetitive [GlcA–GalNAc] for CS and [GlcA–GlcNAc] for HS. Epimerization from CS to DS (i.e., [IdoA–GalNAc] backbone) and from HS to heparin (i.e., [IdoA–GlcNAc] backbone) could occur. The GAG chains are also modified by sulfate groups. Each pathway involves specific glycosyltransferases, sulfotransferases, and epimerases. For KS GAG chain elongation, there are three types of initiations leading to KS-I, KS-II, and KS-III GAG chains, which are composed of a polylactosamine backbone [GlcNAc–Gal]. Enzyme abbreviations: XYLT: Xylosyltransferase; B4GALT7: β-4-galcatosyltransferase7; B3GALT6: β-3-galactosyltransferase6; B3GAT3: β-3-glucuronyltransferase3; CSGALNACT: chondroitin sulfate *N*-acetylgalactosaminyltransferase; CHSY: chondroitin synthase complex; CHST: chondroitin sulfotransferase; DS epi: dermatan sulfate epimerase; U2ST: uronic acid-2-sulfotransferase; D4ST: dermatan-4-sulfotransferase; EXT: exostosin family protein; EXTL: exostosin-like; NDST: *N*-deacetylase/*N*-sulfotransferase; HSST: heparan sulfotransferase; HS epimerase: heparan sulfate epimerase.

**Figure 2 genes-12-01654-f002:**
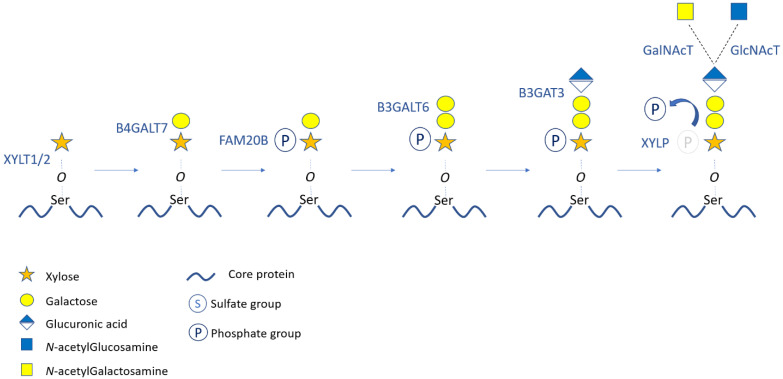
Biosynthesis and modifications of the tetrasaccharide linkage region. The successive steps of the initiating tetrasaccharide biosynthesis and modifications are represented with the corresponding enzymes. Transfer of the first aminosugar of CS or HS chains is also illustrated. The transient phosphorylation of the xylose occurs after the first Gal addition and is removed before the first aminosugar transfer. Enzyme abbreviations: XYLT: Xylosyltransferase; B4GALT7: β-4-galactosyltransferase7; FAM20B: Family with sequence similarity 20B; B3GALT6: β-3-galactosyltransferase6; B3GAT3: β-3-glucuronyltransferase3; XYLP: xylose phosphatase; GalNAcT: N-acetylgalactosaminyltransferase; GlcNAcT: N-acetylglucosaminyltransferase.

**Figure 3 genes-12-01654-f003:**
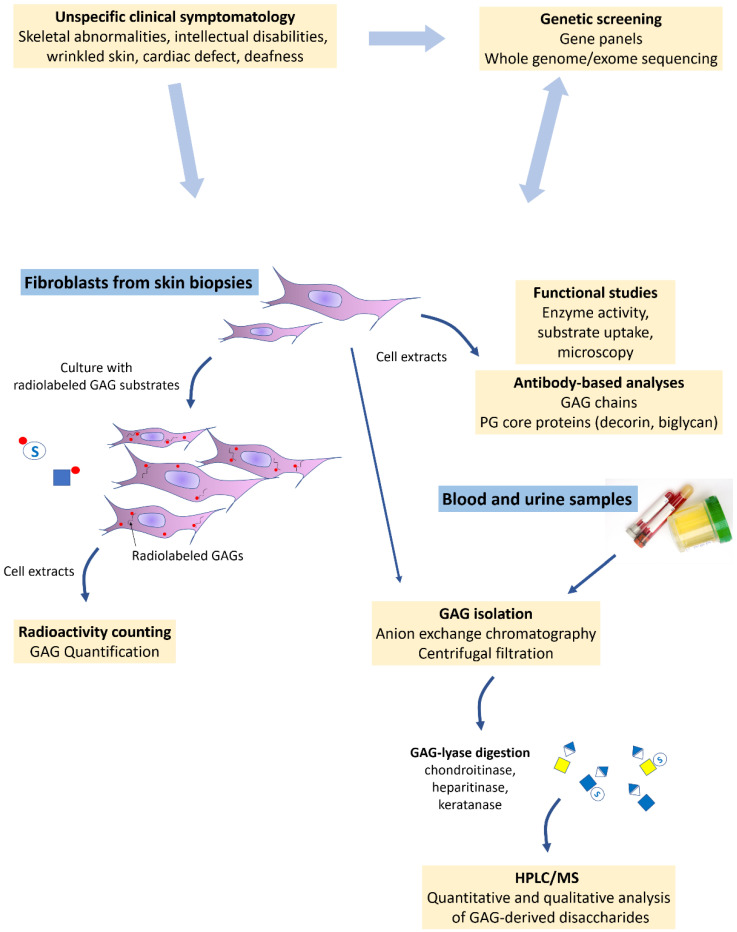
Schematic of current PG-IMD diagnosis strategy. Once PG defect is suspected clinically, genetic screening and GAG assessment are performed using patient’s samples. Whole genome/exome sequencing allows the identification of possibly pathogenic variants in PG metabolism-associated genes. Fibroblasts from skin biopsies are cultured with radiolabeled substrates and the measured radioactivity in cell extracts reflects PG biosynthesis capacities. Fibroblasts are also used for functional analyses measuring the activity of the suspected defective enzyme or transporter, while microscopy could detect protein mislocalization and/or abnormal Golgi morphology. Antibody-based techniques such as Western blot and flow cytometry are useful for highlighting impaired PG biosynthesis by targeting GAG chains or PG core proteins such as decorin and biglycan. As from blood and urine, GAGs could be assessed by analyzing purified and enzymatically released disaccharides using HPLC/MS. Altogether, these laboratory analyses either orient towards a genetic deficiency or confirm the causality of the suspected gene variant.

**Figure 4 genes-12-01654-f004:**
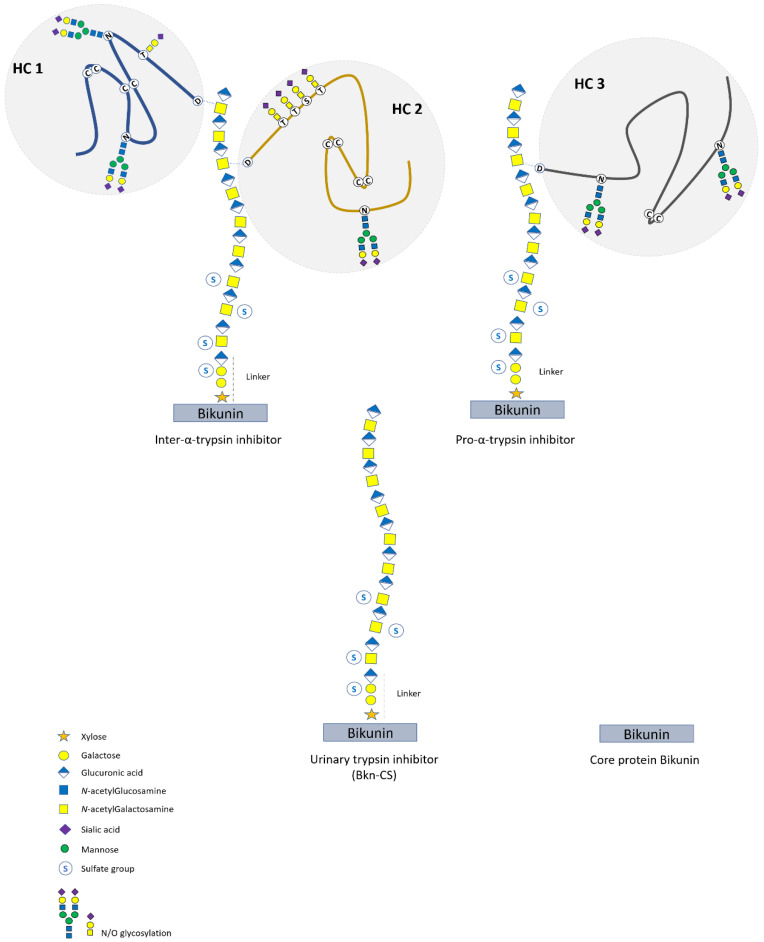
Schematic structure of bikunin isoforms (IAIP). The heavy forms ITI and PαI result from the esterification of the glycoproteins HC1, HC2, and HC3 with the CS chain of the bikunin (Bkn) core protein. The light forms correspond to UTI (Bkn–CS) and the core protein Bkn. The CS chain consists of 15 +/−3 [GlcA–GalNAc] disaccharide units and is sulfated at several GalNAc residues and at the second Gal residue of the linker region.

**Figure 5 genes-12-01654-f005:**
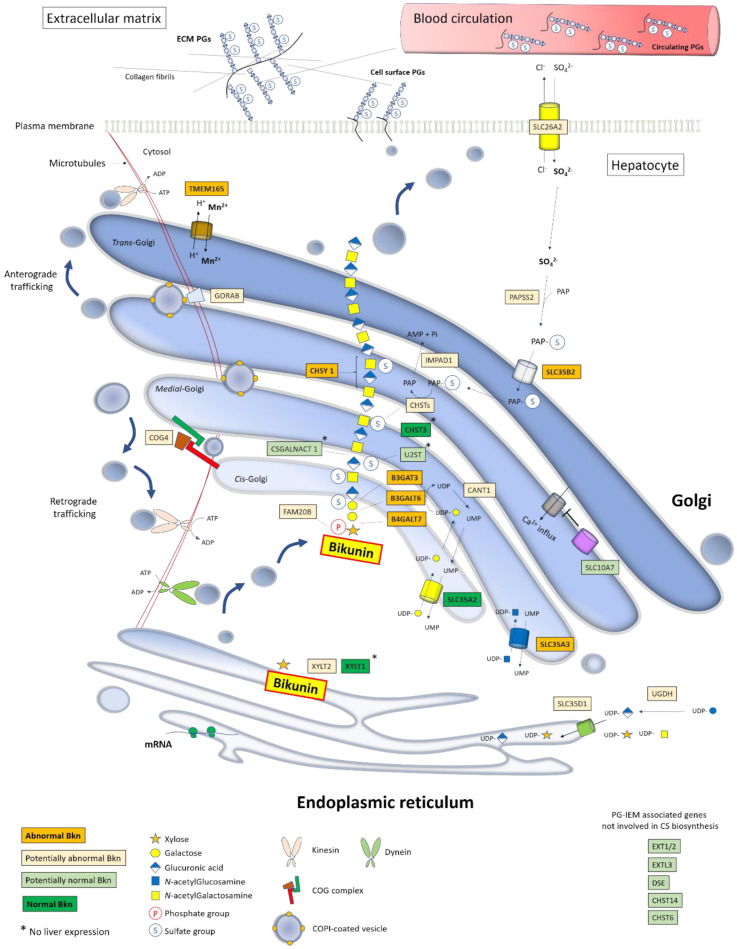
Schematic view of Bkn–CS biosynthesis into the secretory pathway and impact of PG-IMD gene deficiencies on Bkn analysis profile. The function of each PG-IMD-associated gene is depicted together with the Bkn–CS elongation, modifications, and sorting within the secretory pathway in hepatocytes. The latter include an ER-to-Golgi anterograde vesicular flow along microtubules as ensured by dynein molecular motors while retrograde trafficking within the Golgi involves tethering factors (COG complex) and Rab-GTPase-associated proteins such as GORAB. Additionally, retrograde transport to the ER involves kinesins. Golgi homeostasis is maintained by ionic equilibrium (H^+^, Na^+^, Mn^2+^, Ca^2+^) involving ion transporters such as TMEM165. UDP-sugars and PAPS are synthesized in the cytosol by specific enzymes and their uptake within the ER/Golgi lumen is ensured by nucleotide sugar transporters and PAPS transporters, respectively. Nascent PGs are sorted in the TGN and packed into cargo vesicles addressed through kinesin-dependent post-Golgi traffic to the plasma membrane and then to the ECM and/or to blood after exocytosis. Deficiencies in the highlighted genes lead to PG-IMD syndromes, the screening of which could benefit from serum Bkn analysis. Deep and light orange boxes represent gene deficiencies leading to abnormal (already investigated) and potentially abnormal (not yet investigated) Bkn–CS biosynthesis, respectively. Deep and light green boxes include genes whose deficiency results in normal (already investigated) and potentially normal (not yet investigated) Bkn profiles, respectively.

## Data Availability

Not applicable.

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
