# Peer review of "Inherited Proteoglycan Biosynthesis Defects—Current Laboratory Tools and Bikunin as a Promising Blood Biomarker"

_genes, 2021, doi:10.3390/genes12111654_

Round 1
Reviewer 1 Report
The authors described the mini-review of potential biomarkers for inherited proteoglycan biosynthesis, thereby allowing for easier diagnosis and development as well as assessment of the treatment(s). This mini-review may be important and informative for researchers of not only glycobiology field but also other research fields. However, the reviewer felt that the current state of the manuscript might be required for minor modifications as described below.
Thank you very much for allowing me to consider the work.
Sincerely,
Major point:
1) Mucopolysaccharidoses (MPS) are well known a catabolic disease of glycosaminoglycan (GAG). This disease can be also measured the urinary and/or blood GAGs. Thus, the reviewer feels that MPS may be briefly included in the “Introduction” section and discussed in the main text.
2) Bikunin as a potential biomarker for proteoglycan-deficiency has been described in this manuscript. Thus, the reviewer felt that the methodology and/or schema of GAG analysis should be illustrated as a new Figure in the manuscript.
Minor point :
1) Fig. 1:
Figs A and B should be swapped.
The authors described the DS and heparin with branching from CS and HS, respectively. This description might be confused for non-expert audiences.
The authors should utilize the symbols of glucosamine (GlcN), but not N-acetyl-glucosamine (GlcNAc), in case of GlcN of the heparin chain, based on the Symbol Nomenclature for Glycans (SNFG; https://www.ncbi.nlm.nih.gov/glycans/snfg.html ; Glycobiology 25: 1323-1324, 2015; Glycobiology 29:620-624, 2019). Furthermore, the symbol color of iduronic acid is wrong.
“N-acetylGlucosamine” --- > N-acetylglucosamine (N in italic) (Figs 2 and 3 also should be corrected)
“N-acetylGalactosamine” --- > N-acetylgalactosamine (N in italic) (Figs 2 and 3 also should be corrected)
“KS-1”, KS-2”, “KS-3” --- > “KS-I”, KS-II”, “KS-III”, respectively
“-O-Ser/Thr” --- > “O” in italic
The symbol for GlcN should be described in the left lower.
2) The reviewer felt that many abbreviations were utilized in the manuscript, even though they were only or twice.
3) Line 30: “hyaluronic acid” --- > “hyaluronan”
4) Line 61: “[GlcA-Gal-Gal-Xyl]” --- > “[glucuronic acid-galactose-galactose-xylose (GlcA-Gal-Gal-Xyl)]”
5) Lines 63, 64: “[IdoA-GlcNAc] for heparin” --- > “[IdoA-GlcN] for heparin”
6) Line 64: “[GlcNAc-Gal]n” --- > “n” in subscript
7) Line 70: “GA” --- > “Golgi apparatus” (GA may not be common)
8) Line 70: “(ERGIC)” --- > delete
9) Line 71: “cis” --- > in italic
10) Line 75: “2-O phosphorylated” --- > “O” in italic
11) Line 83: “N-acetyl” --- > “N” in italic
12) Line 87: In addition to ref 18, the following reference should be cited.
Uyama T, Kitagawa H, Tanaka J, Tamura J, Ogawa T, Sugahara K. Molecular cloning and expression of a second chondroitin N-acetylgalactosaminyltransferase involved in the initiation and elongation of chondroitin/dermatan sulfate. J Biol Chem. 2003 Jan 31;278(5):3072-8.
13) Line 90: The original articles should be cited instead of the review article, Ref. #19. Because the original articles should be cited in the review, and better for audiences.
Ex)
Kitagawa, H., Uyama, T., and Sugahara, K. (2001). Molecular cloning and expression of a human chondroitin synthase. J. Biol. Chem. 276, 38721-38726.
Kitagawa, H., Izumikawa, T., Uyama, T., and Sugahara, K. (2003). Molecular cloning of a chondroitin polymerizing factor that cooperates with chondroitin synthase for chondroitin polymerization. J. Biol. Chem. 278, 23666-23671.
Izumikawa, T., Uyama, T., Okuura, Y., Sugahara, K., and Kitagawa, H. (2007). Involvement of chondroitin sulfate synthase-3 (chondroitin synthase-2) in chondroitin polymerization through its interaction with chondroitin synthase-1 or chondroitin-polymerizing factor. Biochem. J. 403, 545-552.
Izumikawa, T., Koike, T., Shiozawa, S., Sugahara, K., Tamura, J., and Kitagawa, H. (2008). Identification of chondroitin sulfate glucuronyltransferase as chondroitin synthase-3 involved in chondroitin polymerization: chondroitin polymerization is achieved by multiple enzyme complexes consisting of chondroitin synthase family members. J. Biol. Chem. 283, 11396-11406.
In addition, ChSy, ChPF GalNAcT might be better in all capital. Because the authors utilize other protein names in capital.
14) Lines 92, 93: “EXTL-1, EXTL-2, EXTL-3” --- > “EXTL1, EXTL2, EXTL3”
15) Lines 92, 93: The reviewer felt that the following references should be cited instead of Refs. 20, 21.
Ex)
Lind, T., Tufaro, F., McCormick, C., Lindahl, U., and Lidholt, K. (1998). The putative tumor suppressors EXT1 and EXT2 are glycosyltransferases required for the biosynthesis of heparan sulfate. J. Biol. Chem. 273, 26265-26268.
McCormick, C., Leduc, Y., Martindale, D., Mattison, K., Esford, L. E., Dyer, A. P., and Tufaro, F. (1998). The putative tumour suppressor EXT1 alters the expression of cell-surface heparan sulfate. Nat. Genet. 19, 158-161.
Kim, B. T., Kitagawa, H., Tamura, J., Saito, T., Kusche-Gullberg, M., Lindahl, U., and Sugahara, K. (2001). Human tumor suppressor EXT gene family members EXTL1 and EXTL3 encode a1,4-N-acetylglucosaminyltransferases that likely are involved in heparan sulfate/heparin biosynthesis. Proc. Natl. Acad. Sci. U. S. A. 98, 7176-7181.
Kitagawa, H., Shimakawa, H., and Sugahara K. (1999). The tumor suppressor EXT-like gene EXTL2 encodes an a1, 4-N-acetylhexosaminyltransferase that transfers N-acetylgalactosamine and N-acetylglucosamine to the common glycosaminoglycan-protein linkage region. The key enzyme for the chain initiation of heparan sulfate. J. Biol. Chem. 274, 13933-13937.
16) Line 98: “B3GlcNAc-T7” --- > “B3GlcNAcT7”
17) Line 99: The reviewer felt that the following references should be cited instead of Ref. 22.
Ex)
Seko, A., Dohmae, N., Takio, K., and Yamashita, K. (2003). b1,4-Galactosyltransferase (b4GalT)-IV is specific for GlcNAc 6-O-sulfate. b4GalT-IV acts on keratan sulfate-related glycans and a precursor glycan of 6-sulfosialyl-Lewis X. J. Biol. Chem. 278, 9150-9158.
Seko, A., and Yamashita, K. (2004). b1,3-N-Acetylglucosaminyltransferase-7 (b3Gn-T7) acts efficiently on keratan sulfate-related glycans. FEBS Lett. 556, 216-220.
Kitayama, K., Hayashida, Y., Nishida, K., and Akama, T. O. (2007). Enzymes responsible for synthesis of corneal keratan sulfate glycosaminoglycans. J. Biol. Chem. 282, 30085-30096.
18) Legend for Fig.1 (Lines 101-119):
“KS-1”, KS-2”, “KS-3” --- > “KS-I”, KS-II”, “KS-III”, respectively
“b-3-glucuronic acid transferase3” --- > “b-3-glucuronyltransferase3”
“ChSy” --- > “CHSY” (also in the Figure)
“N-deacetylase/N-sulfotransferase” --- > “N” in italic
17) Line 123: The reviewer felt that the following references should be cited instead of Ref. 23.
Ex)
Yamauchi, S., Mita, S., Matsubara, T., Fukuta, M., Habuchi, H., Kimata, K., and Habuchi, O. (2000) Molecular cloning and expression of chondroitin 4-sulfotransferase. J. Biol. Chem. 275, 8975-8981.
Hiraoka, N., Nakagawa, H., Ong, E., Akama, T. O., Fukuda, M. N., and Fukuda, M. (2000). Molecular cloning and expression of two distinct human chondroitin 4-O-sulfotransferases that belong to the HNK-1 sulfotransferase gene family. J. Biol. Chem. 275, 20188-20196.
Kang, H. G., Evers, M. R., Xia, G., Baenziger, J. U, and Schachner, M. (2002). Molecular cloning and characterization of chondroitin-4-O-sulfotransferase-3. A novel member of the HNK-1 family of sulfotransferases. J. Biol. Chem. 277, 34766-34772.
Fukuta, M., Uchimura, K., Nakashima, K., Kato, M., Kimata, K., Shinomura, T., and Habuchi, O. (1995). Molecular cloning and expression of chick chondrocyte chondroitin 6-sulfotransferase. J. Biol. Chem. 270, 18575-18580. doi: 10.1074/jbc.270.31.18575.
18) Lines 123-5: “GlcA residues can also be sulfated by the uronic acid sulfotransfer- 123 ase (U2ST)” and “Furthermore, varying proportions of GlcA residues of the CS chains can be 124 converted to IdoA by the DS epimerases (DSE)-1 and -2 to form DS chains.” --- > The reviewer felt that the following references should be cited.
Ex)
Kobayashi, M., Sugumaran, G., Liu, J., Shworak, N. W., Silbert, J. E., and Rosenberg, R. D. (1999) Molecular cloning and characterization of a human uronyl 2-sulfotransferase that sulfates iduronyl and glucuronyl residues in dermatan/chondroitin sulfate. J. Biol. Chem. 274, 10474-80.
Maccarana, M., Olander, B., Malmström, J., Tiedemann, K., Aebersold, R., Lindahl, U., Li, J.P., and Malmström, A. (2006). Biosynthesis of dermatan sulfate: chondroitin-glucuronate C5-epimerase is identical to SART2. J. Biol. Chem. 281, 11560-11568.
Pacheco, B., Malmström, A., and Maccarana, M. (2009). Two dermatan sulfate epimerases form iduronic acid domains in dermatan sulfate. J. Biol. Chem. 284, 9788-9795.
19) Line 125-6: “Sulfations oc- 125 cur on both GalNAc and IdoA residues via D4ST (CHST14) and U2ST, respectively [24]” --- > The Ref. 24 were not suitable for this sentence. Thus, the reviewer felt that the following references should be cited instead of Ref. 24.
Ex)
Evers, M. R., Xia, G., Kang, H. G., Schachner, M., and Baenziger, J. U. (2001). Molecular cloning and characterization of a dermatan-specific N-acetylgalactosamine 4-O-sulfotransferase. J. Biol. Chem. 276, 36344-36353.
Kobayashi, M., Sugumaran, G., Liu, J., Shworak, N. W., Silbert, J. E., and Rosenberg, R. D. (1999) Molecular cloning and characterization of a human uronyl 2-sulfotransferase that sulfates iduronyl and glucuronyl residues in dermatan/chondroitin sulfate. J. Biol. Chem. 274, 10474-80.
20) Lines 127-9: “For HSPGs, several GlcNAc residues undergo N-deacetylation and subsequent N-sul- 127 fations catalyzed by bifunctional enzymes named GlcNAc N-deacetylase/N-sulfotransfer- 128 ases (NDSTs) [25].” --- > ”N” in italic. Furthermore, the reviewer felt that the following references should be cited instead of Ref. 25.
Ex)
Hashimoto, Y.; Orellana, A.; Gil, G.; Hirschberg, C. B. J. Biol. Chem. 1992, 267, 15744. Molecular cloning and expression of rat liver N-heparan sulfate sulfotransferase.
Eriksson, I.; Sandbäck, D.; Ek, B.; Lindahl, U.; Kjellén, L. J. Biol. Chem. 1994, 269, 10438. cDNA cloning and sequencing of mouse mastocytoma glucosaminyl N-deacetylase/N-sulfotransferase, an enzyme involved in the biosynthesis of heparin.
21) Lines 129-30: “The resulting sulfoglucosamine (GlcNS) residues can undergo an ad- 129 ditional 6-O sulfation via a 6-O sulfotransferase to form GlcNS,6S.” --- > ”O” in italic. Furthermore, the reviewer felt that the following references should be cited.
Ex)
Habuchi, H.; Tanaka, M.; Habuchi, O.; Yoshida, K.; Suzuki, H.; Ban, K.; Kimata, K. J. Biol. Chem. 2000, 275, 2859. The Occurrence of Three Isoforms of Heparan Sulfate 6-O-Sulfotransferase Having Different Specificities for Hexuronic Acid Adjacent to the Targeted N-Sulfoglucosamine.
22) Lines 130-132: “Additionally, for the formation of heparin, some GlcA residues are converted by the HS-epimerase to IdoA 131 residues which are further sulfated [26].” --- > The reviewer felt that the following references should be cited instead of Ref. 26.
Ex)
Li, J. –P.; Hagner-McWhirter, Å.; Kjellén, L.; Palgi, J.; Jalkanen, M.; Lindahl, U. J. Biol. Chem. 1997, 272, 28158. Biosynthesis of Heparin/Heparan Sulfate: cDNA CLONING AND EXPRESSION OF D-GLUCURONYL C5-EPIMERASE FROM BOVINE LUNG.
23) Lines 132-4: “For KSPGs, sulfations occur on both GalNAc and Gal residues composing the KS chain and are catalyzed by Gal6ST (CHST6) and GlcNAc6ST [27].” --- > “For KSPGs, sulfations occur on both GalNAc and Gal residues composing the KS chain and are catalyzed by Gal6ST (CHST1) and GlcNAc6ST (CHST5 and/or CHST6) [Refs].” Furthermore, the reviewer felt that the following references should be cited instead of Ref. 27.
Ex)
Fukuta, M., Inazawa, J., Torii, T., Tsuzuki, K., Shimada, E., and Habuchi, O. (1997). Molecular cloning and characterization of human keratan sulfate Gal-6-sulfotransferase. J. Biol. Chem. 272, 32321-32328.
Akama, T. O., Nakayama, J., Nishida, K., Hiraoka, N., Suzuki, M., McAuliffe, J., Hindsgaul, O., Fukuda, M., and Fukuda. M. N. (2001). Human corneal GlcNac 6-O-sulfotransferase and mouse intestinal GlcNac 6-O-sulfotransferase both produce keratan sulfate. J. Biol. Chem. 276, 16271-16278.
Akama, T. O., Misra, A. K., Hindsgaul, O., and Fukuda, M. N. (2002). Enzymatic synthesis in vitro of the disulfated disaccharide unit of corneal keratan sulfate. J. Biol. Chem. 277, 42505-42513.
24) Lines 138, 223, 270: “GA” --- > “Golgi apparatus”
25) Line 157: “SO42-“ --- > “4 and 2 as well as minus” should be in subscript and superscript, respectively.
26) Lines 172, 184, 199, 314: “hyaluronic acid” --- > “HA”
“PG inherited metabolic disorders” --- > deleted
27) Lines 213, 216: The refs “50, 51, 53, and 54” was not first report. The reviewer felt that the following references should be cited instead of these refs.
Ex)
Munns, C. F., Fahiminiya, S., Poudel, N., Munteanu, M. C., Majewski, J., Sillence, D. O., Metcalf, J. P., Biggin, A., Glorieux, F., Fassier, F., Rauch, F., and Hinsdale, M. E. (2015). Homozygosity for frameshift mutations in XYLT2 result in a spondylo-ocular syndrome with bone fragility, cataracts, and hearing defects. Am. J. Hum. Genet. 96, 971-978.
Almeida, R., Levery, S. B., Mandel, U., Kresse, H., Schwientek, T., Bennett, E. P., and Clausen, H. (1999). Cloning and expression of a proteoglycan UDP-galactose:beta-xylose b1,4-galactosyltransferase I. A seventh member of the human b4-galactosyltransferase gene family. J. Biol. Chem. 274, 26165-26171.
Baasanjav, S., Al-Gazali, L., Hashiguchi, T., Mizumoto, S., Fischer, B., Horn, D., Seelow, D., Bassam R. Aziz, A.S., Langer, R., Saleh, A.A.H., Becker, C., Nürnberg, G., Cantagrel, V., Gleeson, J. G., Gomez, D., Michel, J.B., Stricker, S., Lindner, T.H., Nürnberg, P., Sugahara, K, Mundlos, S., and Hoffmann, K. (2011). Faulty initiation of proteoglycan synthesis causes cardiac and joint defects. Am. J. Hum. Genet. 89, 15-27.
Ahn, J., Lüdecke, H. J., Lindow, S., Horton, W. A., Lee, B., Wagner, M. J., Horsthemke, B., and Wells, D. E. (1995). Cloning of the putative tumour suppressor gene for hereditary multiple exostoses (EXT1). Nat. Genet. 11, 137-143.
Stickens, D., Clines, G., Burbee, D., Ramos, P., Thomas, S., Hogue, D., Hecht, J. T., Lovett, M., and Evans, G. A. (1996). The EXT2 multiple exostoses gene defines a family of putative tumour suppressor genes. Nat. Genet. 14, 25-32.
28) Line 225: “Mn2+ and Ca2+“ --- > “2+” should be in superscript.
29) Line 244: “35S and 3H“ --- > “35 and 3” should be in superscript.
30) Lines 260-1: “[IdoA-GalNAc4S] --- > “[IdoA-GalNAc4-O-sulfate]
31) Line 267: The ref #71 was redundant with ref. # 69.
32) Line 306: “trans-“ --- > in italic
33) Figure 2:
The symbol for mannose should be described in the left lower.
“N-acetylGlucosamine” --- > N-acetylglucosamine (N in italic)
“N-acetylGalactosamine” --- > N-acetylgalactosamine (N in italic
34) Line 349: “CHSY-1” --- > “CHSY1”
35) Figure 3:
“N-acetylGlucosamine” --- > N-acetylglucosamine (N in italic)
“N-acetylGalactosamine” --- > N-acetylgalactosamine (N in italic
36) Legend for Figure 3:
“H+, Na+, Mn2+, Ca2+” --- > “+ and 2+” should be in superscript.
“NSTs and PAPSTs” --- > “nucleotide sugar transporters and PAPS transporters”
37) Abbreviations:
Line 431: “Family with sequence similarity member 20-B” --- > delete the hyphen
Lines 435, 437: “N-acetygalactosamine and N-acetylglucosamine” --- > “N” in italic
Line 439: “hyaluronic acid” --- > hyaluronan
Line 450: “N-deacetylase/N-sulfotransferases” --- > “N” in italic
Line 453: “phospho-adenosine phospho-sulfate” --- > “3’-phosphoadenosine 5’-phosphosulfate”
Line 460: “trans” -- > in italic
Line 462: “uronic acid sulfotransferase” --- > “ulonyl 2-O-sulfotransferase” or “ulonosyl 2-O-sulfotransferase”
Line 468: “XYLT1/2” --- > “XYLT”
38) Supplementary Table S2:
The cited references, 18, 19, 21, 23, 26, 28, 29, 30, 32, 33, 38, 46, 47 should be corrected as described above.
The disorder caused by mutation in Fam20B may not be “type 2”. Thus, “type2” should be deleted.
“Ser-O-xyl(P)-gal” --- > “Gal-Xyl(2-O-phosphate)-O-Ser” (“O” in italic)
“CSGALNACT-1, CHSY-1, CHST-12, CHST-13, CANT-1, COG-4” --- > deleted the phyphens
“D4ST1 (CHST14) --- > “CHST14 (D4ST1)”
“U2ST” --- > “UST” (because the gene name is UST)
“GA” in SLC35A2, COG4 ---- > Golgi apparatus
“UGDH” --- > in italic
39) References in main text and Supplementary information
The format for cited references should be unified and confirmed author’s guideline.
Reviewer 2 Report
Dear editor,
Thank you very much for the opportunity to review this manuscript.
Proteoglycans (PG) are a ubiquitous family of macromolecules consisting of specific core proteins bearing one or more O- or N-linked sulfated glycosaminoglycan (GAG) chains, with key structural and functional roles in many tissues that can be found in both plasma and urine. GAGs are very heterogeneous polysaccharides in terms of the type of repeating disaccharide unit, chain length, charge density, degree and pattern of sulfation, degree of epimerization, and physicochemical properties, being responsible for most of the numerous biological functions of PGs.
As summarized in Supplementary Table 2 by Haouari et al., there are many inherited mutations causing defects in PGs biosynthesis that result in severe diseases including skeletal abnormalities associated to multiorgan impairments. Developing screening methods for their diagnosis based on blood or urine samples is desirable. In this respect, bikunin represents a potential biomarker.
This mini-review is interesting and well written. I have no major concerns about the publication of this paper. Two minor point should be addressed by the authors:
Rows 274-5. This sentence should be rephrased. The description of these methods is not clear, as, generally, GAGs are purified by chromatography followed by specific lyases treatment for structural analysis (see also ref 73), whereas in ref 74 no chromatographic step was performed before MS analysis.
Rows 347-363. The application of 1D or 2D electrophoresis followed by WB analyses seems to represent a promising screening tool. These technologies, especially 2DE followed by WB, are quite time consuming and do not provide clear structural information on GAG chains. Authors should discuss more in details pros and cons.
